# BepiColombo mission confirms stagnation region of Venus and reveals its large extent

M. Persson [1], S. Aizawa[1], N. André[1], S. Barabash[2], Y. Saito[3], Y. Harada [4], D. Heyner[5], S. Orsini [6], A. Fedorov [1], C. Mazelle [1], Y. Futaana[2], L. Z. Hadid[7], M. Volwerk[8], G. Collinson[9], B. Sanchez-Cano [10], A. Barthe[1], E. Penou[1], S. Yokota [11], V. Génot [1], J. A. Sauvaud[1], D. Delcourt[7], M. Fraenz [12], R. Modolo [13], A. Milillo [6], H.-U. Auster[5], I. Richter [5], J. Z. D. Mieth[5], P. Louarn[1], C. J. Owen [14], T. S. Horbury[15], K. Asamura[3], S. Matsuda [16], H. Nilsson[2], M. Wieser [2], T. Alberti[6], A. Varsani [8], V. Mangano[6], A. Mura [6], H. Lichtenegger[8], G. Laky[8], H. Jeszenszky[8], K. Masunaga [3], C. Signoles[1], M. Rojo[1] & G. Murakami[3]

The second Venus flyby of the BepiColombo mission offer a unique opportunity to make a complete tour of one of the few gas-dynamics dominated interaction regions between the supersonic solar wind and a Solar System object. The spacecraft pass through the full Venusian magnetosheath following the plasma streamlines, and cross the subsolar stagnation region during very stable solar wind conditions as observed upstream by the neighboring Solar Orbiter mission. These rare multipoint synergistic observations and stable conditions experimentally confirm what was previously predicted for the barely-explored stagnation region close to solar minimum. Here, we show that this region has a large extend, up to an altitude of 1900 km, and the estimated low energy transfer near the subsolar point confirm that the atmosphere of Venus, despite being non-magnetized and less conductive due to lower ultraviolet flux at solar minimum, is capable of withstanding the solar wind under low dynamic pressure.

The presence of a planetary intrinsic magnetic field and/or an ionosphere defines the nature of the interaction between a Solar System object and the solar wind, and what effects this interaction has on the atmospheric evolution. Venus has a crushingly thick atmosphere but, albeit an Earth-like planet, lacks both a global magnetic field, such as Earth's, and crustal magnetic fields, such as Mars[1], and therefore only its ionosphere is interacting with the solar wind[2]. The interaction induces currents in the conductive ionosphere and forms an induced

[1]Institut de Recherche en Astrophysique et Planétologie, Centre National de la Recherche Scientifique, Centre National d'Etudes Spatiales, Université Paul Sabatier—Toulouse III, Toulouse, France. [2]Swedish Institute of Space Physics, Kiruna, Sweden. [3]Institute of Space and Astronautical Science, Japan Aerospace Exploration Agency, Kyoto, Japan. [4]Department of Geophysics, Graduate School of Science, Kyoto University, Kyoto, Japan. [5]Institute for Geophysics and Extraterrestrial Physics, Technische Universität Braunschweig, Braunschweig, Germany. [6]Institute of Space Astrophysics and Planetology, Istituto Nazionale di Astrofisica, Rome, Italy. [7]Laboratoire de Physique des Plasmas (LPP), Centre National de la Recherche Scientifique, Observatoire de Paris, Sorbonne Université, Université Paris Saclay, École Polytechnique, Institut Polytechnique de Paris, Paris, France. [8]Space Research Institute, Austrian Academy of Sciences, Graz, Austria. [9]National Aeronautic and Space Administration, Goddard Space Flight Center, Greenbelt, MD, USA. [10]School of Physics and Astronomy, University of Leicester, Leicester, UK. [11]Department of Earth and Space Science, Graduate School of Science, Osaka University, Osaka, Japan. [12]Max-Planck-Institute for Solar System Research, Göttingen, Germany. [13]Laboratoire Atmosphères, Milieux, Observations Spatiales, Institut Pierre Simon Laplace, Université Versailles Saint Quentin en Yvelines, Université Paris-Saclay, Université Pierre Marie Curie, Centre National de la Recherche Scientifique, Guyancourt, France. [14]Mullard Space Science Laboratory, University College London, Holmbury St. Mary, UK. [15]Imperial College London, South Kensington Campus, London, UK. [16]Graduate School of Natural Science and Technology, Kanazawa University, Kanazawa, Japan. ✉e-mail: moa.persson@irap.omp.eu

magnetosphere, which becomes an obstacle to the solar wind[3]. As the solar wind passes through the detached bow shock it is decelerated and a magnetosheath is formed around the obstacle. At the subsolar point of the magnetosheath, the solar wind dynamic pressure and ionospheric thermal pressures are the largest and the solar wind is decelerated to a stagnated flow, i.e., reaches a very low bulk speed and a high temperature. Measurements in the Venusian magnetosheath allow us to study the pure theoretically-predicted gas-dynamic interaction between the supersonic solar wind and a conductive ionosphere[4]. Given its unique characteristics, the Venusian magnetosheath is also a perfect natural laboratory in our Solar System for investigating processes such as the energy transfer from the solar wind to the ionosphere of non-magnetized bodies[5], and the properties of the stagnated flow near the subsolar point[6] in an environment free of complications associated with magnetic fields of planetary origin.

Several gas-dynamic models have been developed to describe the gas-dynamics dominated magnetosheath of Venus[7–11], of which the overall results were confirmed and improved on by the available in situ plasma measurements from several previous successful missions. The Pioneer Venus Orbiter (PVO[12]) mission, which orbited Venus in 1978–1992, could complete a full solar cycle of measurements. Although its magnetometer gave a detailed view of the magnetic fields in the Venusian magnetosheath[13], its plasma particle instruments did not have a high enough time resolution and energy range to provide comprehensive particle characteristics. In addition, due to a raise of its periapsis after the first few years, it could not sample the subsolar magnetosheath during solar minimum conditions (Fig. 1A). The Venus Express (VEx[14]) mission, which orbited Venus in 2006–2014, provided more details on the Venusian magnetosheath. However, due to its highly elliptical polar orbit it could not perform in situ measurements in the subsolar magnetosheath (Fig. 1A). In addition, no previous mission has had a flyby

trajectory that sampled the Venusian subsolar magnetosheath, which is the critical region that ultimately defines the solar wind-planet interaction. Therefore, this textbook example of a pure gas-dynamics interaction region between the solar wind and a non-magnetized object have not yet been experimentally confirmed by in situ plasma measurements.

Here, we show that the BepiColombo Venus flyby on Aug 10, 2021 (Fig. 1B) provided an opportunity to experimentally confirm the sub-regions of the subsolar magnetosheath previously predicted by models[7–11]. The observations confirm the presence of the stagnation region and show that it can extend to an altitude of 1900 km, which support that there is a limited entry of and energy transfer from the solar wind to the ionosphere of Venus during low solar wind dynamic pressure conditions.

## Results

### A flyby through the Venusian magnetosheath
During its 2-h flyby of Venus on the 10th of August 2021, the dual-spacecraft BepiColombo mission[15] passed through the Venusian magnetosheath from the nightside down to the almost unexplored subsolar region roughly following along the streamlines (Fig. 1B). At this time, it consisted of the Mercury Magnetosphere Orbiter[16] (MMO, now called Mio) and the Mercury Planetary Orbiter (MPO) integrated into a stacked configuration together with the Mercury Transfer Module (MTM) and the Mio sunshield and interface structure (MOSIF). See Fig. 2 for an exploded view of the spacecraft composition. The flyby gives us access to this interaction region, showing us a complete and almost instantaneous picture of the different subregions of the magnetosheath; where the solar wind is heated and significantly decelerated, where it is deflected around Venus, and finally where it is accelerated up to almost solar wind speeds again along the flank[3,4]. A summary of the BepiColombo plasma measurements is shown in Fig. 3. The tour occurred during the rare observational

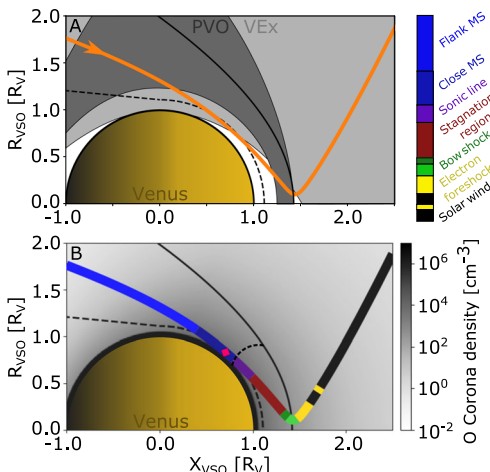

**Fig. 1 | The trajectory of BepiColombo 2nd Venus flyby in the cylindrical Venus-Solar-Orbital (VSO) coordinates.** In VSO the x-axis points along the Venus-Sun line, the y-axis in the Venus anti-orbital direction and z-axis completes the orthogonal system; here $R = \sqrt{y^2 + z^2}$. **A** BepiColombo's trajectory (orange) compared with the orbit coverage of the missions that have crossed the subsolar magnetosheath, VEx (light gray) and PVO (dark gray). The orange arrow head indicates the trajectory direction. **B** The trajectory of BepiColombo divided into different colors depending on the plasma region traversed, as identified from the measurements, where the colorbar defines these regions. Most noteworthy is the maroon color part of the trajectory, which is the barely-explored stagnation region, a subregion of the subsolar magnetosheath. The red diamond shows the closest approach. The background gray colormap presents the expected oxygen (O) corona density[27]. The expected location of the sonic line is indicated by the smaller dashed line[7]. In both panels the averaged bow shock and ion composition boundary are indicated by the curved and dashed black lines[20], respectively.

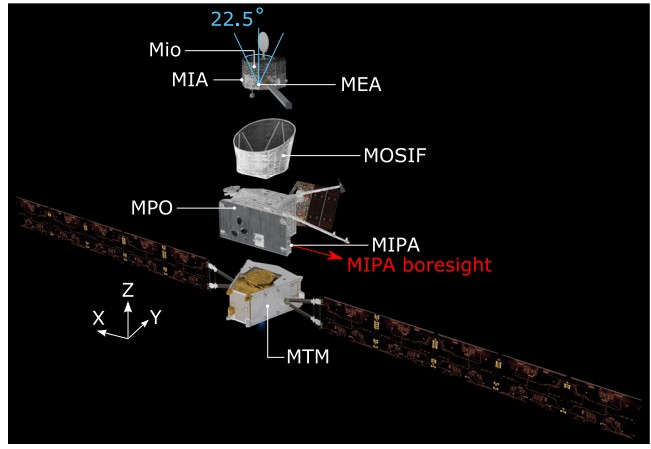

**Fig. 2 | Exploded view of the composition of the BepiColombo spacecraft during cruise phase.** The different spacecraft parts and their names are marked with white lines, including the location of MEA, MIA and MIPA. The two pixels of the MEA instrument FoV which point outside of MOSIF are marked by the blue angles. The few pixels that point outside of MOSIF for MIA are similar in shape and direction to MEA's, albeit being placed on a different location on Mio (MIA's pixels are also shown in Fig. 5)[37]. The boresights of MEA and MIA are along the +Z axis of the inset coordinate system in white (the MPO spacecraft frame). The boresight direction of MIPA is marked with the red arrow and is along the −X axis (at a 90° angle from MEA and MIA boresights). The total MIPA FoV is along the −X axis and covers approximately a wide cone of 80°[49], where the defined pixels used during this Venus flyby are shown in Fig. 5. Note that all acronyms are defined in the main text. Image by ESA/ATG medialab, adapted by adding names and indicators of instruments under the license CC BY-SA 3.0 IGO.

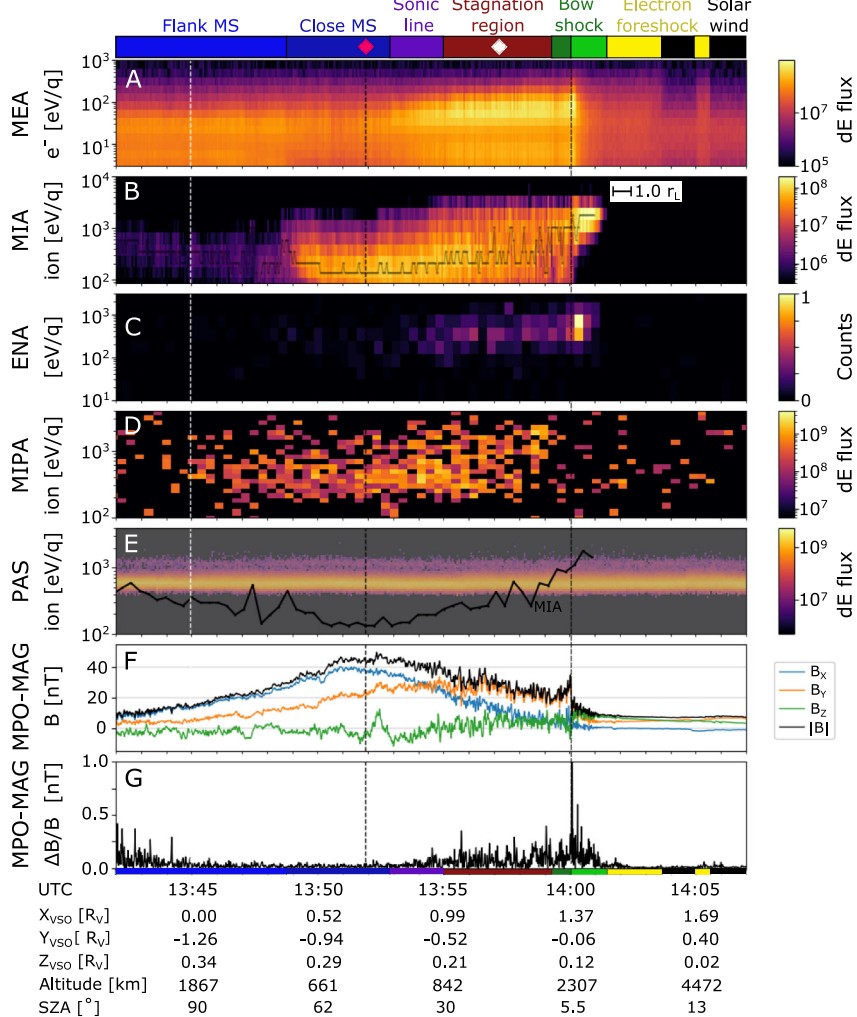

**Fig. 3 | Electron, ion and magnetic field measurements made by the three spacecraft MPO, Mio, and Solar Orbiter during BepiColombo's 2$^{nd}$ Venus flyby on August 10, 2021.** Energy-time spectrogram of omni-directional (**A**) Mio/MPPE/MEA differential energy flux (dE flux) [cm$^{-2}$ s$^{-1}$ eV/eV][37,47], (**B**) Mio/MPPE/MIA differential energy flux [cm$^{-2}$ s$^{-1}$ eV/eV][37] (with black line showing the peak energy bin), (**C**) Mio/MPPE/ENA normalized counts[37], which nominally measures energetic neutral atoms but here measures neutrals originating from protons neutralized by the interaction with the spacecraft structures and thus operates as a very sensitive monitor of proton fluxes, (**D**) MPO/SERENA/MIPA differential energy flux [cm$^{-2}$ s$^{-1}$ eV/eV][49], (**E**) Solar Orbiter/PAS differential energy flux [cm$^{-2}$ s$^{-1}$ eV/eV][18] (shifted by 1 h, and slightly shaded) with the peak energy bin from MIA (black line, smoothed for clarity) overplotted for comparison, (**F**) the magnetic field measured by MPO-MAG[50] in VSO coordinates, and (**G**) the variations of the magnetic field measured by MPO-MAG. All ion measurements are shown integrated over mass. The terminator crossing is indicated by the vertical dashed white line, and the closest approach of 550 km altitude by the vertical black dashed line. The inset in B shows the approximate length of one gyroradius (r$_L$) calculated from the solar wind conditions near the bow shock. The colorbar on top shows the regions identified from the changes in the plasma parameters measured by the different instruments. The bow shock is differentiated into two green shades for the ramp and ion foot (light green) and the over- and undershoot (dark green). The subsolar magnetosheath (MS) is differentiated into two subregions:[7] the stagnation region (brown) and the sonic line (purple). The red diamond shows the closest approach, and the white diamond shows the time stamp for the temperature calculations in Fig. 7. The colors are also shown along the trajectory in Fig. 1B. Estimation of the bow shock parameters from observations gives the angle between the shock normal and the magnetic field (θ$_{Bn}$) of 82°.

opportunity when another spacecraft, Solar Orbiter, performed a flyby of Venus the day before[17]. Therefore, Solar Orbiter was located upstream of Venus (Fig. 4), at about 200 Venus radii from the planet, along the same Parker Spiral arm, and could obtain complementary solar wind and magnetic field measurements[18,19]. The multi-spacecraft observational configuration was further complemented by favorable upstream conditions: the solar wind and the interplanetary magnetic field were stable during the entire BepiColombo flyby as observed by Solar Orbiter (Fig. 4). These stable conditions provided an opportunity to investigate purely the spatial variability of the magnetosheath without the interference of temporal variabilities imposed by fluctuations in the solar wind.

## Quasi-perpendicular bow shock crossing

Following the flow of solar wind protons along the expected plasma streamlines as they skim Venus (and so going in the direction opposite to the orbit of BepiColombo, backwards in time), we first encounter signatures of the planet's presence downstream from the electron foreshock visible in observations by the Mercury Electron Analyzer (MEA) onboard Mio at 14:05 UT (Fig. 3). We cross the quasi-perpendicular bow shock near the subsolar point of Venus (Fig. 3, 13:59-14:01 UT), when the angle between the interplanetary magnetic field direction and the solar wind flow direction was about 90°, as observed by the magnetometer (MPO-MAG) onboard MPO (Fig. 2F). The crossing location matches the expected distance from Venus, as found from VEx observations[20] (Fig. 1B). The solar wind is not apparent

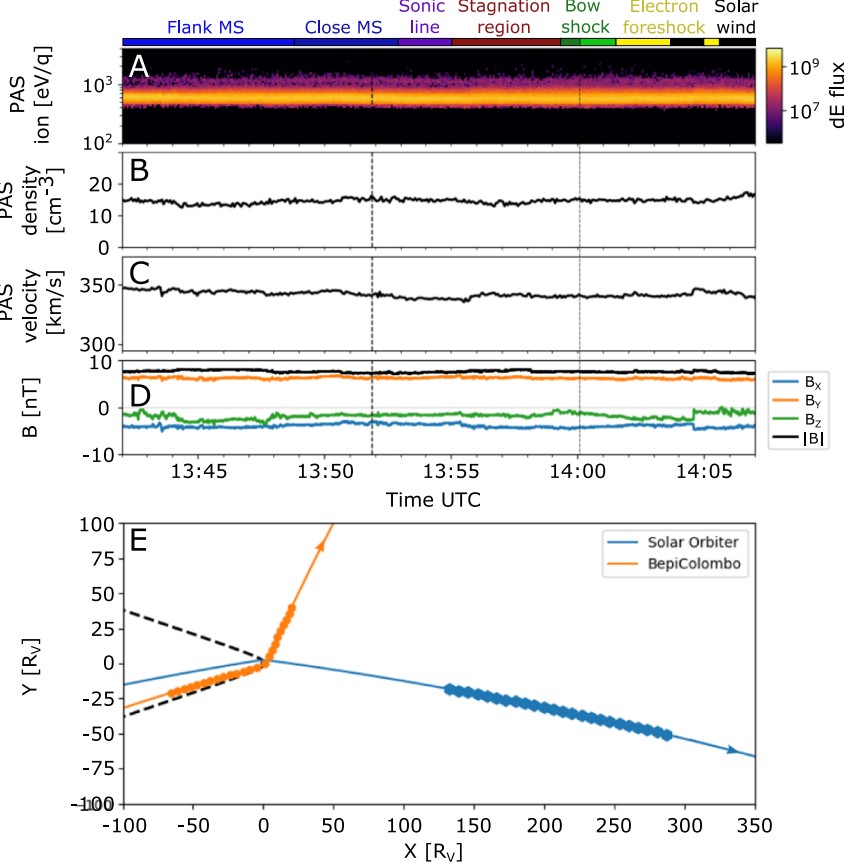

**Fig. 4 | Measurements of the solar wind plasma and magnetic field made by Solar Orbiter upstream of Venus on August, 10 2021, shifted to the BepiColombo position (1 h shift). A** Time-Energy spectrogram of PAS proton energy differential flux [cm$^{-2}$ s$^{-1}$ eV/eV], (**B**) proton density, (**C**) proton speed, (**D**) magnetic field components in the VSO coordinate system. (**E**) Location of the Solar Orbiter (blue, one symbol per hour) and BepiColombo (orange, one symbol per hour) in the x–y plane of the VSO coordinate system (see Fig. 1 caption), during the Venus flyby of BepiColombo on August 10, 2021. The directions of the orbits are indicated by the arrows. The average bow shock is indicated by the dashed black line[20]. The colors on the top of the figure separates the plasma regions identified from the BepiColombo measurements.

in the ion measurements by Mercury Ion Analyzer (MIA), onboard Mio, and Miniature Ion Precipitation Analyzer (MIPA), onboard MPO, upstream of the bow shock due to a combination of the attitude of BepiColombo, which gives a 90° angle between the instrument boresights and the solar wind flow direction, and the limited field-of-view (FoV) of the instruments onboard Mio during the cruise phase (Figs. 2, 3B, D). In the foot of the bow shock (at 14:01 UT), some of the solar wind protons are specularly reflected by the cross-shock electric field. These protons perform half a gyration, upstream from the shock ramp, around the magnetic field, while being accelerated along the bow shock surface by the motional electric field, before they can enter through the bow shock ramp (14:00 UT). This proton population was observed out to approximately one gyroradius (about 400 km) from the bow shock ramp by both MIA and Energetic Neutrals Analyzer (ENA, which is designed to measure energetic neutral atoms but here measures neutrals originating from protons neutralized by the interaction with MOSIF) onboard Mio (Fig. 3B, C). However, it was not observed by MIPA onboard MPO (Fig. 3D), due to the spacecraft attitude and the large difference in FoV between MIA and ENA compared to MIPA, which have no overlap and boresights separated by ~90° (Figs. 2, 5). The gyrating protons at the bow shock are here observed to be accelerated to on average 3.2 times the energy of the solar wind, found by the comparison between the solar wind energy measured by Proton-Alpha Sensor (PAS) onboard Solar Orbiter, shifted in time to the location of BepiColombo, and the peak energy measured by MIA onboard Mio (Fig. 3E). This is within the expected acceleration, where

the reflected protons can reach up to nine times the solar wind energy, but only after a 180° gyration in the planetary frame[21–23].

As the solar wind beam and the gyrating proton populations pass through the shock, they are heated. The heating typically occurs on the scale of a gyroradius[24]. In the transition region the two proton populations are separated in velocity space, until they become fully mixed and look almost thermalized[24–26]. The separation of the MIA and MIPA FoV boresights by 90° (Figs. 2, 5) results in that the gyrating population is observable for MIA (Fig. 3B) already from 14:01 UT, while for MIPA only at 13:59 UT when this population becomes sufficiently hot to reach its FoV (Fig. 5A). This leads to the apparent disappearance of the signal in the MIPA records between 13:58 and 14:01, while MIA (and ENA) was still observing the gyrating population (Fig. 3B, D).

### Stagnation region
Closer to Venus, the flow is expected to slow down considerably (at 13:55–13:58 UT). Due to the stacked configuration of BepiColombo during the cruise phase and the limited FoV (Fig. 2), the real ion bulk velocity and thermal speed from both MIPA and MIA cannot be accurately calculated from the available partial ion distributions. However, MIPA observes a clear change in the measurements; from the observed heated gyrating protons just downstream of the bow shock (Fig. 5A), to entering a region where the protons have a larger spread in direction and a lower count rate (Fig. 5B), to entering a region with a proton flow more aligned with the expected flow along the obstacle boundary (see next paragraph) and a higher count rate (Fig. 5C). The large spread and

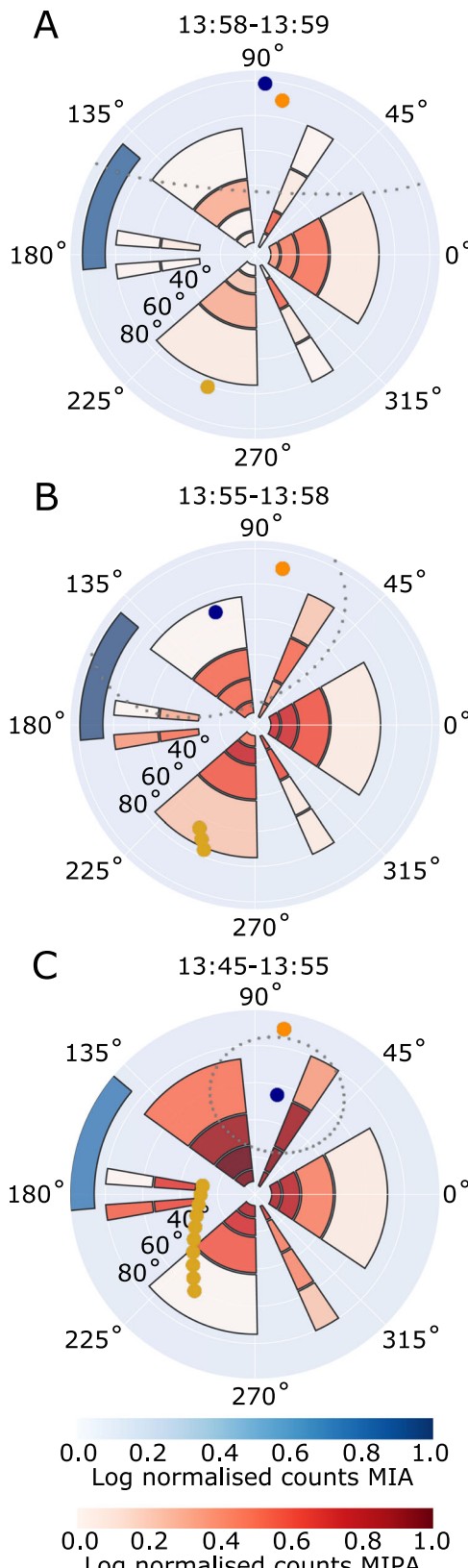

**Fig. 5 | The direction of the measured proton counts in MIPA and MIA angular field-of-view.** The azimuth and radial directions represent the azimuth and elevation angles in the MIPA instrument coordinate system. Each plot shows the log normalized counts in MIPA (red) and MIA (blue) instrument pixels, integrated over time and energy for (**A**) 13:58-13:59, (**B**) 13:55–13:58, and (**C**) 13:45–13:55. The yellow dots show the direction of Venus' center (one per minute), and the orange dot shows the direction of the Sun. The measurements can be compared to the results from the LatHyS global hybrid model (Fig. 6), where we can extract the expected direction of the average bulk velocity (here shown as a blue dot) and the expected width of the proton distribution (shown as gray dots, calculated from the average thermal speed), at the location of BepiColombo in each time range. Note that the LatHyS bulk speed is lower than the LatHyS thermal speed in (**A**, **B**) (see Fig. 6C), and that the peak of the measured distribution in (**C**) is located within 20–30° of the bulk flow direction from LatHyS (blue dot) with significantly overlapping distributions, which both indicate a good match between the measurements and the simulations.

decelerated due to the presence of the Venusian induced magnetosphere. The large amplitude of the magnetic field fluctuations (Fig. 3G) is consistent with a higher plasma density fluctuation, as expected in a compressed high-density region. Whereas the location of the stagnation region may change with respect to external parameters, such as extreme ultraviolet radiation, upstream solar wind speed, or Mach number[6], our unique, in situ observations of the stagnation region show that it extends to altitudes of at least 1900 km near the subsolar region (Fig. 1B) under the conditions prevalent during the BepiColombo flyby. This indicates a large region of stagnated flow during an interval with low solar wind pressure.

### Sonic line
Following the flow that is deflected around Venus, the spacecraft ends up in another subsolar magnetosheath subregion characterized by mixed particle distributions around 13:53–13:55 UT. Here, the properties of almost all plasma parameters show signatures of changes. The electron and proton energies decrease (Fig. 3A–D), the magnetic field fluctuations decrease in amplitude (Fig. 3G), and the direction of the flow changes (Fig. 5). The flow in this region is approximately aligned with the obstacle boundary (i.e., the Venusian induced magnetosphere boundary) and is within the MIPA FoV (Fig. 5C). These changes occur near the expected location of the sonic line, a transition region predicted by gas-dynamic models[7], where the flow transitions from subsonic to supersonic speeds, and beyond which the plasma flow is expected to revert to almost solar wind speeds further downstream[8].

### Closest approach
As the spacecraft is nearing the closest approach of Venus (at 13:52 UT, red diamond in Fig. 1B), a depletion in the thermal electron flux is observed, which is close to the region where the hot oxygen corona density is predicted to peak along the BepiColombo trajectory[27] (Fig. 1B). In addition, the peak of the magnetic field magnitude is observed in sync with the depletion, which, together with the decrease in the magnetic field fluctuations, indicates an entering of the magnetic pile up boundary, a region mostly void of solar wind protons[28–33]. Furthermore, the spacecraft might have skimmed the induced magnetosphere boundary[3,34], where the dominating species changes from solar wind origin to planetary origin, but is unlikely to have entered the induced magnetosphere. Whereas the small peak in the magnetic field $B_Z$ component (at 13:52 UT) could signify an encounter with the boundary current layer, there is no evidence of a complete crossing in the available magnetic field and plasma data.

### Flank magnetosheath
After the closest encounter of Venus induced magnetosphere, a change is again detected in the environment, mostly caused by the increase in altitude, which shows that the spacecraft moves away from

the lower count rate at 13:55–13:58 UT (Fig. 5B) implicate that the protons are more thermalised here compared to at 13:45–13:55 UT (Fig. 5C), which indicate that the thermal speed of the protons is higher than their bulk speed. Thus, these observations indicate that the spacecraft reached the stagnation region, a subregion of the Venusian subsolar magnetosheath, where the solar wind is significantly

the induced magnetosphere boundary back into the undisturbed flank magnetosheath flow (<13:49 UT). The lower temperature and higher bulk speed of the protons in this region, in addition to the change back to a perpendicular angle between the flow direction and the ion instruments' boresights, are the main reasons for the decrease in the observed fluxes by the ion instruments. The Mass Spectrum Analyzer (MSA) onboard Mio, dedicated to composition analysis, made measurements up until 13:49 UT, where a safe mode automatic procedure turned it off due to an exceedingly high ion flux entering the instrument.

### Back into the solar wind flow

Around 13:42 UT (near the left edge of Fig. 3), inside the flank magnetosheath of Venus, the protons are observed to be accelerated up to almost the solar wind energy measured by PAS onboard Solar Orbiter (Fig. 3E, where the black line returns to almost the same energy as the pristine solar wind), although their distribution still exhibits the expected signatures of being shocked, heated and slightly slowed down. These signatures prevail until the spacecraft leaves the magnetosheath, passing out into the solar wind through the bow shock, at around 12:00 UT and a distance of ~8 $R_V$ (1 $R_V$ is one Venusian radius, ~6052 km) from Venus.

## Discussion

The sampling of the Venusian magnetosheath done by BepiColombo during its second Venus flyby provides us with detailed insight into the structure and properties of the gas-dynamic dominated interaction region between the solar wind and Venus, at near solar minimum for conditions of low solar wind pressure. As the sampling was made during stable solar wind conditions, as measured by the upstream Solar Orbiter, we had the opportunity to investigate the pure spatial variations, without fluctuations typically induced by the temporal variations of the solar wind. The observations near the subsolar magnetosheath show us a passage through the almost unexplored stagnation region, where the solar wind is significantly heated and slowed down. This is confirmed by a comparison with the output from a global hybrid model (LatHyS[35,36], Fig. 6). Aizawa et al.[35] showed that the model is well constrained by the upstream solar wind conditions measured by Solar Orbiter (Fig. 4) and that the validity of the model output is confirmed through the good match between its magnetic field components and the MPO-MAG measured magnetic field components (Fig 6[35]). Here, we use their model output to focus on interpreting the BepiColombo data. The solar wind proton speeds in the subsolar magnetosheath from the model confirm the interpretations from the measurements: the spacecraft entered the stagnation region, where the proton bulk speed is seen to be lower than the proton thermal speed (Figs. 5B, 6C).

In addition to the omni-directional electron differential energy flux provided by MEA (Fig. 3A), the instrument also obtains one full 3D scan (with the flux separated into its angular pixels), integrated over 4 s, every 10 min during the flyby[37]. One such scan is obtained inside the identified stagnation region, which allows us to investigate the electron population in this region in more detail. From fitting a Maxwellian distribution to the electron energy spectra we can thus show that the in situ measured electron temperature in this region is 34 eV (Fig. 7A). The same method applied to the MIA instrument provides an average ion temperature of 280 eV during near 13:57 UT (Fig. 7B). The electron temperature is lower than the previously expected temperatures of around 90–140 eV[38]. The lower electron temperature limits the efficiency of the electron impact ionization processes in the subsolar magnetosheath[39], which is equal to, or even higher than, the photo-ionization process in the subsolar magnetosheath[38]. The lower ionization, together with the lower density of the oxygen corona during near solar minimum conditions[40], indicates a lower pickup ion density in the subsolar magnetosheath, both compared to the solar maximum

conditions and compared to previous assumptions from models[38]. This is important, as the planetary ion density determines the level of mass loading of the solar wind and thus the assumed outer boundary of the solar wind void, which is assumed to be the magnetic pile up boundary (MPB)[41,42].

The subsolar magnetosheath is a key region for understanding how much energy is transferred from the solar wind to the ionosphere through its inner boundary. This boundary, the MPB, was shown to not allow the penetration of the solar wind during solar maximum conditions[30] and in the flank magnetosheath during solar minimum[28]. However, the subsolar MPB has not yet been investigated during solar minimum conditions. Here, BepiColombo provided an opportunity to investigate the energy transfer at the subsolar point during near solar minimum conditions. As there is a pressure balance at the interface between the Venusian ionosphere and the solar wind, we may calculate the magnetic field strength of the MPB: The dynamic pressure of the solar wind is converted to thermal pressure in the stagnation region, then to magnetic pressure in the MPB, and finally into thermal pressure in the ionosphere[3]. From the Solar Orbiter measurements we find that the solar wind dynamic pressure was 1.4 nPa during the flyby, which gives a magnetic field strength of the MPB in the subsolar point of ~55 nT[30] (or similarly we find approximately the same number from assuming a $\cos^2$ (solar zenith angle) relationship[43] of the peak magnetic field that was measured in the flank magnetosheath at closest approach). The measured proton temperature (Tp) together with the maximum magnetic field strength provides an average proton gyro-radius of about 30 km (100 km if assuming a much higher Tp of 3 keV). This is significantly smaller than both the previously measured thickness of the MPB (around 300 km[30]) as well as the altitude of the MPB (around 600 km[20,30]) and its distance to the ionosphere (peak altitude at ~150 km[44]). Even when assuming that the boundary altitude is lower, due to the near solar minimum time period, a significant penetration of gyrating protons will not be observed. This indicates that the solar wind cannot interact directly with the ionospheric particles and thus cannot transfer energy directly through Coulomb collisions. Therefore, the BepiColombo measurements show that the inner boundary of the Venusian magnetosheath is efficient in excluding the solar wind also at near solar minimum and low dynamic pressure conditions, at least when the IMF displays a large angle with the solar wind flow direction.

These unique observations at Venus by plasma instruments onboard the two spacecraft of the BepiColombo mission, put into context by observations of stable external solar wind conditions directly upstream by a third spacecraft, Solar Orbiter, indicate that the Venusian ionosphere is also strong in the subsolar point during solar minimum, enforced by the observed large stagnation region, the low electron temperature, and the small proton gyroradius. This confirms the limited entry of and energy transfer from the solar wind to the planetary ionosphere in the absence of intrinsic or crustal magnetic field. This is an important finding as it relates to the connection between magnetic fields and atmospheric escape due to solar wind erosion, which is important for understanding a planet's habitability[45]. In addition, it shows the important additions plasma measurements from single flybys can make for Solar System objects, such as Venus.

## Methods

### Instrumentation

The observational data used in this study comes from the BepiColombo and Solar Orbiter missions. The BepiColombo mission is traveling in a stacked configuration during the cruise phase and is composed of both the MPO (European Space Agency, ESA) and Mio (Japan Aerospace Exploration Agency, JAXA) spacecraft.

Onboard the Mio spacecraft we used the Mercury Plasma Particle Experiment (MPPE) instrument consortium. MPPE/MEA consists of two

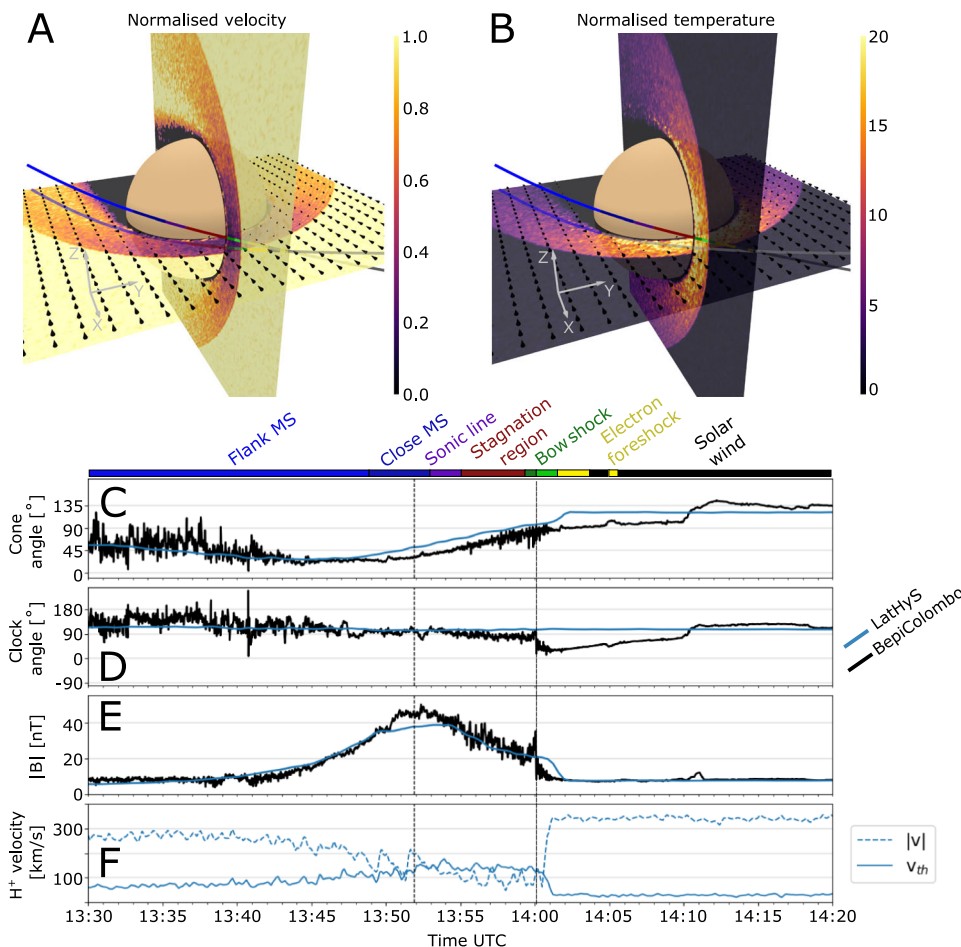

**Fig. 6 | LatHyS simulation of Venus during the BepiColombo flyby, constrained by Solar Orbiter upstream solar wind measurements. A** Proton speeds, normalized by the upstream solar wind speed of 338 km/s, and (**B**) proton temperature, normalized by the upstream solar wind temperature of 10.7 eV, presented in two planes of the VSO coordinate system (see small inset gray coordinate arrows) from a LatHyS model[35] hybrid simulation constrained by upstream solar wind plasma measurements obtained by Solar Orbiter/PAS[18] and Solar Orbiter/MAG[19] instruments obtained during 12:40–13:00 UTC: density 14.7 cm⁻³, dynamic pressure 2.8 nPa, plasma beta (proton) 1.1, Mach number 7.7, solar EUV flux 10.7 value 70 and magnetic field **B** = [−3.87, 6.2, −1.8]. The black arrows represent the flow directions. The colored line shows the BepiColombo trajectory, with the same colors as used in Fig. 3 (also shown on top of **C**), which represent the separation of different sub-regions found by the measurements. The gray line shows the flyby trajectory projected onto the x–y plane. The simulation results confirm that the trajectory of BepiColombo was almost aligned with the obstacle boundary (see also Fig. 5C). Comparison between the measured magnetic field by MPO-MAG (in black) and the magnetic field from the LatHyS simulation (in blue) against time (in UT) along the trajectory of BepiColombo for (**C**) cone angle, (**D**) clock angle and (**E**) magnetic field magnitude. **F** The proton bulk speed (solid line) and thermal speed (dashed line) along the BepiColombo trajectory. The closest approach and the bow shock crossing are marked with dashed vertical black lines.

sensors (here we used MEA1) which nominally measures the phase space density of low energy electrons between 0.003–26 keV[37]. MPPE/MIA nominally measures the phase space density of low energy ions of 0.015–29 keV[37]. MPPE/MSA measures the mass separated phase space density of low energy ions of 0.001–38 keV[37]. MPPE/ENA measures mass separated energetic neutral atoms between 0.01 and 3.3 keV, however, due to the stacked configuration some FoV pixels will also measure ions neutralized by the spacecraft structures[37].

Onboard the MPO spacecraft we used the Search for Exospheric Refilling and Emitted Natural Abundances (SERENA) instrument consortium and the MPO-MAG magnetometer. SERENA/MIPA measures the phase space density of ions within 0.015–15 keV[46]. MPO-MAG is composed of two tri-axial fluxgate magnetometers mounted on a boom of 2.9 m (0.8 m apart), which measure magnetic fields of ±2048 nT at up to 128 Hz[47].

Onboard Solar Orbiter we used the PAS instrument, part of the Solar Wind Analyser (SWA) suite and the MAG magnetometer. SWA/PAS measures the phase space density of ions within 0.20–20 keV[18]. MAG is composed of two tri-axial fluxgate magnetometers, which

measures the local magnetic field of up to ±60,000 nT at up to 128 Hz[19].

## Temperature fitting

The temperature is calculated through assuming a drifting Maxwellian distribution in the phase space density of the electron (MEA) and ion (MIA) measurements

$$f(v) = N\sqrt{\frac{m_p}{2\pi T}}e^{-\frac{m_p(v - v_s)^2}{2T}} \qquad (1)$$

where $N$ is the density, $m_p$ is the proton mass, $T$ is the temperature, $v$ is the speed and $v_s$ is the drifting speed of the Maxwellian distribution.

## LatHyS global hybrid simulation

In this study the measurements are compared with the output from the LatHyS global hybrid simulation developed for the Venusian environment[35]. The simulation solves the interaction between the solar wind and the Venusian environment by treating

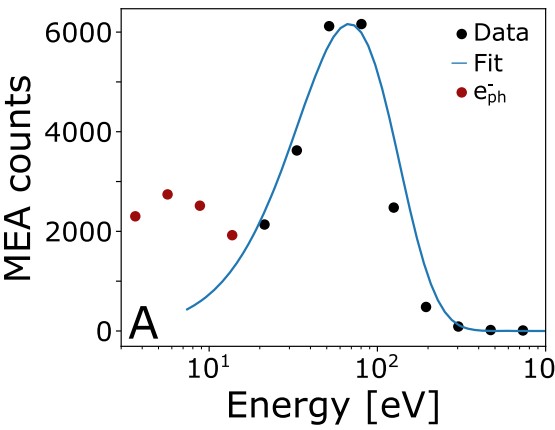

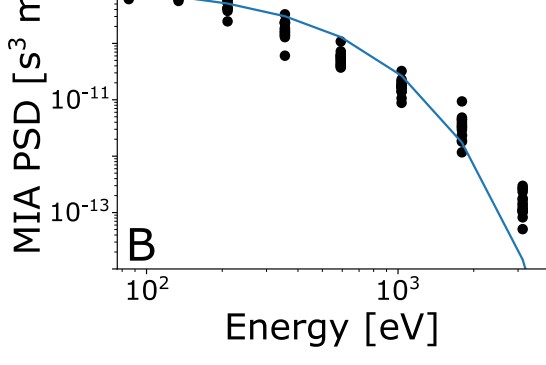

**Fig. 7 | Temperature calculations from fitting to the measured energy distributions of ions and electrons.** Maxwellian distribution fits to (**A**) electron counts measured by MEA and (**B**) proton phase space density (PSD) measured by MIA. The black dots show the measurement and the blue line shows the fitted distribution. The electron distribution was measured at 13:57 UT, and the 15 ion distributions used for the fit were measured between 13:57 and 13:58 UT. The fitted distributions, assuming a close to zero V spacecraft potential, give a temperature of 34 eV for the electrons (which is also strengthened by the approximation that the temperature equals the peak energy, here about 70 eV, divided by two[51]) and 280 eV for the ions. The electron measurements below 10 eV (red dots) correspond to a population of photoelectrons ($e_{ph}^{-}$) attached to Mio due to spacecraft charging and is not included in the fitting.

ions as particles and electrons as mass-less charge-neutralizing fluid. The behavior of the ions is obtained by solving the Lorentz equation of motion, and the magnetic and electric fields are obtained by solving Maxwell's equations. The ionosphere is self-consistently created from the ionization of the exosphere, through photoionization, charge exchange and electron impact ionization.

## Data availability

All data used to support the conclusions in this study are presented in the main paper. The data presented here are available for download in the Zenodo database at https://doi.org/10.5281/zenodo.7297140[48]. The Mio/MPPE, MPO/SERENA and MPO-MAG observational data can be requested from the respective PIs: Y.S. (saito@stp.isas.jaxa.jp), S.O. (stefano.orsini@inaf.it) and D.H. (d.heyner@tu-braunschweig.de), by describing the intent of the use of the data, as discussions with the respective PI is needed for analysing the data, due to the complex configuration and operations of the two attached spacecrafts (Mio and MPO) of the Bepi-Colombo mission during cruise phase. After the proprietary period of 12 months, the BepiColombo mission data analyzed in this study will be available at the ESA-PSA archive https://archives.esac.esa.int/psa/#!Table%20View/BepiColombo=mission as soon as the data products are ready. We used the L2 data of the Solar Orbiter MAG and SWA data in this study, which are publicly available at the Solar Orbiter Archive Repository (https://soar.esac.esa.int/soar/) of the European Space Agency. The simulation data used in this study are available at http://bepi-colombo.irap.omp.eu/documents/PUBLICATIONS/AIZAWA/Bepi_VF2/.

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

## Acknowledgements

The authors acknowledge all members of the BepiColombo and Solar Orbiter missions for their unstinted efforts in making these missions successful. In particular the MEA Team would like to thank Claude Aoustin for managing the technical activities of the instrument development at IRAP. The authors would also like to express their gratitude to J.G. Luhmann at Space Sciences Laboratory, U.C. Berkeley for helpful discussions. French co-authors acknowledge the support of Center National d'Etudes Spatiales (CNES, France) to the BepiColombo and Solar Orbiter mission. BepiColombo is a joint space mission between ESA and JAXA. MPPE is funded by JAXA, CNES, the Center National de la Recherche scientifique (CNRS, France), the Italian Space Agency (ASI). SERENA management are funded by ASI, the Italian National Institute of Astrophysics (INAF), and the ground-based activities by ESA (EXPRO contract). SERENA/PICAM is funded by the Austrian Space Applications Program of the Austrian Research Promotion Agency (FFG), ESA's Program de Développement d'Expériences (PRODEX) and CNES. The Swedish Contribution to SERENA/MIPA and MPPE/ENA is funded by the Swedish National Space Agency (SNSA). Solar Orbiter is a space mission of international collaboration between ESA and NASA, operated by ESA. Solar Orbiter Wind Analyser (SWA) data are derived from scientific sensors which have been designed and created, and operated, under funding provided in numerous contracts from the UK Space Agency (UKSA), the UK Science and Technology Facilities Council (STFC), ASI, CNES, CNRS, the Czech contribution to the ESA PRODEX program, and NASA. Solar orbiter SWA work at UCL/MSSL is currently funded under STFC grants ST/T001356/1 and ST/S000240/1. Solar Orbiter magnetometer operations are funded by the UK Space Agency (grant ST/T001062/1); T.S.H. is supported by STFC grant ST/S000364/1. B.S.-C. acknowledges support through UK-STFC Ernest Rutherford Fellowship ST/V004115/1 and STFC grant for BepiColombo travel ST/V000209/1. DH was supported by the German Ministerium für Wirtschaft und

Energie and the German Zentrum für Luft-und Raumfahrt under contract 50 QW1501. S.A. is funded by the French National Research Agency (ANR) for the TEMPETE (Temporal Evolution of Magnetized Planetary Environments during exTreme Events) project. M.P. is funded by the European Union's Horizon 2020 program under grant agreement No 871149 for Europlanet 2024 RI.

## Author contributions

M.P., S.A., N.A., S.B. conceived the study, analyzed the data, and wrote the initial draft of the paper. All authors have read and provided feedback on the paper. Y.S. is Principal Investigator (PI) of the MPPE consortium. G.M. is the project scientist of BepiColombo Mio for JAXA. S.O. is PI of the SERENA consortium. D.H. is PI of the MPO-MAG. C.J.O. is PI of the SWA consortium. T.S.H. is PI of the Solar Orbiter MAG consortium. B.S.C. is Guest Investigator for the BepiColombo mission. Y.H., A.F., C.M., Y.F., L.Z.H., A.B., E.P., S.Y., V.G., J.A.S., D.D., M.F., R.M., P.L., K.A., S.M., K.M. are co-Is of MPPE. M.V., H.U.A., I.R., J.Z.D.M. are co-Is of MPO-MAG. A.M.I., H.N., M.W., T.A., A.V., V.M., A.M.U., H.L., G.L., H.J. are co-Is of SERENA. A.F. and P.L. are co-Is of SWA. G.C., C.S., M.R. are collaborators.

## Competing interests

The authors declare no competing interests.
