## [Peer Review File · Nature Communications]

REVIEWER COMMENTS

Reviewer #1 (Remarks to the Author):

This is an interesting paper relating to unique data on the Venus-solar wind interaction gathered by BepiColombo and Solar Orbiter during the flyby on 10 August 2021. The noteworthy results are

- observations of the solar wind stagnation region with modern instrumentation
- large extent of the region
- the atmosphere of an unmagnetized planet like Venus can withstand the solar wind for low solar wind dynamic pressure, a finding relevant for extrasolar planet studies

Some of the BepiColombo plasma results were already shown in a related paper (ref 29), but much more interpretation, data and analysis is shown here in detail

The work appears to support the conclusions, but a few clarifications and additions would be helpful:

1. Line 108, figure referred to is Fig 3 not 2
2. Line 111, Fig 2 F not G?
3. Line 129, is $3.2 \times$ solar wind energy expected for gyrating ions (the maximum energy expected is $4 \times$ solar wind in the right frame, as for pickup ions of the same species?)
4. Line 137-139 - this discussion is rather unclear, it would be helped by having the fields of view of MIPA, MIA, MEA clarified, and ideally indicated on Fig 2 to ease the discussion.
5. Line 167-169, It would be helpful when discussing the 'sonic line' to calculate the Mach number from available data - is this possible?
6. Line 177-178 - The induced magnetosphere boundary is mentioned, what would be the evidence for crossing this? This might help non-specialists understand the context and claim here

Review of Persson+, “BepiColombo’s scenic tour at Venus reveals large stagnation region”

The manuscript presents analysis and interpretation of time series plasma observations made by ESA’s BepiColombo spacecraft as it flew by Venus on 10th August 2021. Coincident in time observations made by ESA’s Solar Orbiter spacecraft, which was located near to and upstream of Venus, accompany the BepiColombo observations. The manuscript presents a detailed analysis of the Venusian environment as observed by BepiColombo during the flyby, in particular identifying an extended flow stagnation region at the nose of the planet. While this flow stagnation region has been predicted by various gas dynamic models (as noted in the manuscript), BepiColombo is the first spacecraft to sample the sub-solar magnetosheath region using in-situ measurements and the presented observations and interpretations thus provide new and novel constraints on the flow dynamics of the Venusian magnetosphere. The multi-point measurements obtained by the inclusion of Solar Orbiter measurements upstream of Venus allow the authors to demonstrate that the solar wind is stable during the BepiColombo flyby. Variability observed by BepiColombo is thus likely driven by spatial variability in the Venusian magnetosphere rather than temporal variability that would otherwise be driven by the solar wind and its interaction with Venus.

While the key result of this manuscript confirms the existence of a feature predicted by earlier modeling studies (the flow stagnation region), I believe this manuscript is still very much of great interest to the scientific community. The study confirms the existence of the large stagnation region at the sub-solar magnetosheath via direct observation, a feat unachievable by previous spacecraft data sets due to observation limitations (which are discussed in the study). The combination of BepiColombo with Solar Orbiter measurements provides reliable constraints on the system and further demonstrates the power of multi-point observations for planetary science.

With regards to the novelty of the results in the context of reference 29 (Aizawa et al. (2022)), I believe that the manuscript presents novel analysis and discussion that is not included in reference 29. Reference 29 includes in its abstract the sentence “The simulation confirms that BepiColombo passed through the stagnation region of Venus, which supports the results obtained by data analysis.” I did not find any further discussion (or indeed mention) of the flow stagnation region in the main text of reference 29. Reference 29 focuses on the verification of the hybrid model when run at Venus, via comparison with Venus Express and BepiColombo observations (the BepiColombo data are for the same time range analyzed in this manuscript). The “scientific focus” of reference 29 was on characterizing the ion escape rates, and their differences, during the simulation runs. In contrast, this manuscript presents a very focused discussion of the plasma environment encountered by BepiColombo during the Venus flyby, presenting several new and novel datasets that were not part of previous Venus orbiter missions. As such, I believe this manuscript presents novel discussion not included in reference 29, and that these manuscripts well complement each other.

I do have several minor clarification questions that I believe should be addressed prior to any publication; these are outlined below.

General comment on Figure 3 and accompanying text (lines ~82-141): I found the descriptions and walk through of this figure, and the instrument observation strengths and limitations, to be very thorough and useful.

Line 55: is “peculiarities” the right word here? Can you mention what these are if so? Would “unique characteristics”, or something similar, be more appropriate?

Line 67: While PVO ion measurements were limited in some properties, they still provided the fluid characteristics of the Venus environment. I suggest modifying this sentence to something like “did not have enough time resolution and energy range to provide comprehensive/complete particle characteristics”.

Line 86: Should the Mercury Magnetosphere Orbiter have the acronym “MMO”, not Mio?

Lines 144-159: discussion of the stagnation region as observed by the particle instruments: I found this section difficult to follow, and I suggest rewording it if possible. Can you clarify, is it that in Figure 5B, the ion distribution function (idf) is broad and so you are assuming that the ions have been heated and the flow stagnates? To my uninitiated eye, panels B and C seem quite similar – both have broad idfs. The main difference appears to be that in B, the counts do not overlap with the blue dot, while they do in C. Is this important? Perhaps you can show some sort of line plot with normalized integrated counts as a function of radius in FOV, to show that the idf in B is the broader than in C, if this is the case?

Line 165: Given the field of view limitations discussed previously, what is the uncertainty here that the flow is ~aligned with the obstacle boundary? In Figure 5C, is there a symbol denoting the obstacle boundary? How is the obstacle boundary defined?

Line 167: Text on lines 163-164 states that electron and ion energies decrease, but at line 167 the text states that the flow transitions from subsonic to supersonic. Are these consistent?

Line 168: “beyond which the plasma flow reverts to almost solar wind speeds further downstream” Is this the black line in Figure 3E? I suggest pointing that out if so. Can you comment on the accuracy of this interpretation, given the field of view issues discussed earlier?

Lines 175: At Mars, when spacecraft enter the MPB, there is sudden reduction in high energy (>~70 eV) electrons - that does not seem to be the case here (e.g. Vignes+ 2000). There is also no clear large scale rotation in magnetic field to a draped configuration. Do you still think that this is the MPB ?

Line 202: Is the word “instantaneous” appropriate here? What is the sampling instantaneous with?

Line 212: Would the word “interpretations” be better than “assumptions”?

Figure 2 caption: Are all of the acronyms defined in the text?

Figure 3: does panel B show data for all ion species, or just a particular species? Please clarify this in the manuscript if it's not done so already.

Figure 5: this ties in with my comments about lines 144-159 above. For panel B, is the blue dot where the stagnation region is expected to be (the blue dots are labeled as the expected average bulk flow velocity in the caption)? If so, why are there no counts in the sector overlapping the blue dot, if this is the stagnation region?

References:

EG Vignes, D., Mazelle, C., Rme, H., Acuña, M. H., Connerney, J. E. P., Lin, R. P., ... & Ness, N. F. (2000). The solar wind interaction with Mars: Locations and shapes of the bow shock and the magnetic pile-up boundary from the observations of the MAG/ER Experiment onboard Mars Global Surveyor. *Geophysical Research Letters*, 27(1), 49-52.

Reviewer #3 (Remarks to the Author):

Review by T. E. Cravens

This paper presents a wealth of data from BepiColombo's second encounter with Venus and from the Solar Orbiter upstream of Venus in the solar wind. The paper focuses on the plasma and field properties in the region downstream of the bow shock including the magnetosheath and stagnation regions. Although these regions were studied earlier by the Pioneer Venus and Venus Express missions, the current paper provides a different and more coherent view of these regions. Results from global hybrid simulations of the solar wind interaction with Venus (also shown in Aizawa et al., 2022) are very helpful in putting the BepiColombo data into context. The paper will make an important contribution to our understanding of how the solar wind interacts with non-magnetic planets. I recommend publication with a few rather minor suggestions for improvement as listed below.

- Three regions need to be more carefully defined and distinguished in the paper: magnetosheath, stagnation region, and magnetic pile-up region. Looking at Figures 1 and 3, the brown region labeled stagnation region appears to be subsolar magnetosheath plasma downstream of the shock. And the flow continues to be slow into the purple region where the magnetic field is increasing (magnetic pile-up region).
- The data is clearly presented in the paper, despite its complexity. One possible exception is Figure 5 which shows MIP and MIPA proton distributions. It is difficult to explain such complex data from such instruments in a rather short paper. A few more sentences might help.
- Lines 234 to 255 in the Discussion section puts the data into a broader dynamical context. The statement on magnetic pressure in lines 240-244 can be made even more explicit by stating the magnetic pressure (i.e., 1.2 nPa for 55 nT) and upstream solar wind dynamic pressure $\approx ?$ from Solar Orbiter. A general statement might also help the reader: solar wind dynamic pressure tends to be converted into thermal pressure (shock and stagnation) and then into magnetic pressure (barrier/pile-up region).

REVIEWER COMMENTS

REVIEWER #1 (REMARKS TO THE AUTHOR):

This is an interesting paper relating to unique data on the Venus-solar wind interaction gathered by BepiColombo and Solar Orbiter during the flyby on 10 August 2021. The noteworthy results are

- observations of the solar wind stagnation region with modern instrumentation
- large extent of the region
- the atmosphere of an unmagnetized planet like Venus can withstand the solar wind for low solar wind dynamic pressure, a finding relevant for extrasolar planet studies

Some of the BepiColombo plasma results were already shown in a related paper (ref 29), but much more interpretation, data and analysis is shown here in detail

The work appears to support the conclusions, but a few clarifications and additions would be helpful:

Dear reviewer, thank you for reviewing our manuscript and for the positive and constructive comments. We have revised the manuscript with regards to your comments and provide point-by-point replies below.

1. Line 108, figure referred to is Fig 3 not 2

Changed.

2. Line 111, Fig 2 F not G?

Changed.

3. Line 129, is 3.2 x solar wind energy expected for gyrating ions (the maximum energy expected is 4 x solar wind in the right frame, as for pickup ions of the same species?)

Added a sentence that compares this to the full acceleration of up to nine times the solar wind energy in the planetary frame after a 180° gyration. Since we are only seeing a part of this reflected proton distribution, at a specific angle, we don't expect to see them at their maximum acceleration. Note that the maximum acceleration is higher than that for pickup ions since we are dealing with reflected protons from the core solar wind distribution, which have an important initial velocity in the shock frame, in comparison with the pickup ions that can be considered to be initially at rest in the planetary frame. Added references on this in the text.

4. Line 137-139 - this discussion is rather unclear, it would be helped by having the fields of view of MIPA, MIA, MEA clarified, and ideally indicated on Fig 2 to ease the discussion.

Added a pointer to Fig 2 in the text, and clarified the FoVs of the MIPA, MEA and MIPA in Figure 2 and its caption.

5. Line 167-169, It would be helpful when discussing the 'sonic line' to calculate the Mach number from available data - is this possible?

The limitations on the instruments during cruise phase (as mentioned in the manuscript) makes the calculation of the Mach number difficult. However, the different changes that we do observe in the measurements are a clear indication that we are indeed crossing from one spatial region to the next, which we attribute to the sonic line crossing as predicted by the gas-dynamic models that we are referring to (i.e. ref 7-8: Spreiter & Stahara, 1985, 1992).

6. Line 177-178 - The induced magnetosphere boundary is mentioned, what would be the evidence for crossing this? This might help non-specialists understand the context and claim here

The induced magnetosphere boundary is typically considered to be the boundary between the region dominated by solar wind species and the region dominated by planetary species. Added this to the text for clarification.

REVIEWER #2 (REMARKS TO THE AUTHOR):

Please see attached comments. **(text from pdf pasted below)**

Review of Persson+, “BepiColombo’s scenic tour at Venus reveals large stagnation region”

The manuscript presents analysis and interpretation of time series plasma observations made by ESA’s BepiColombo spacecraft as it flew by Venus on 10th August 2021. Coincident in time observations made by ESA’s Solar Orbiter spacecraft, which was located near to and upstream of Venus, accompany the BepiColombo observations. The manuscript presents a detailed analysis of the Venusian environment as observed by BepiColombo during the flyby, in particular identifying an extended flow stagnation region at the nose of the planet. While this flow stagnation region has been predicted by various gas dynamic models (as noted in the manuscript), BepiColombo is the first spacecraft to sample the sub-solar magnetosheath region using in-situ measurements and the presented observations and interpretations thus provide new and novel constraints on the flow dynamics of the Venusian magnetosphere. The multi- point measurements obtained by the inclusion of Solar Orbiter measurements upstream of Venus allow the authors to demonstrate that the solar wind is stable during the BepiColombo flyby. Variability observed by BepiColombo is thus likely driven by spatial variability in the Venusian magnetosphere rather than temporal variability that would otherwise be driven by the solar wind and it’s interaction with Venus.

While the key result of this manuscript confirms the existence of a feature predicted by earlier modeling studies (the flow stagnation region), I believe this manuscript is still very much of great interest to the scientific community. The study confirms the existence of the large stagnation region at the sub-solar magnetosheath via direct observation, a feat unachievable by previous spacecraft data sets due to observation limitations (which are discussed in the study). The combination of BepiColombo with Solar Orbiter measurements provides reliable constraints on the system and further demonstrates the power of multi-point observations for planetary science.

Dear reviewer, thank you for reviewing our manuscript and providing positive and constructive comments. We have adjusted the manuscript according to your comments and addressed them line-by-line below.

With regards to the novelty of the results in the context of reference 29 (Aizawa et al. (2022)), I believe that the manuscript presents novel analysis and discussion that is not included in reference 29. Reference

29 includes in its abstract the sentence “The simulation confirms that BepiColombo passed through the stagnation region of Venus, which supports the results obtained by data analysis.” I did not find any further discussion (or indeed mention) of the flow stagnation region in the main text of reference 29. Reference 29 focuses on the verification of the hybrid model when run at Venus, via comparison with Venus Express and BepiColombo observations (the BepiColombo data are for the same time range analyzed in this manuscript). The “scientific focus” of reference 29 was on characterizing the ion escape rates, and their differences, during the simulation runs. In contrast, this manuscript presents a very focused discussion of the plasma environment encountered by BepiColombo during the Venus flyby, presenting several new and novel datasets that were not part of previous Venus orbiter missions. As such, I believe this manuscript presents novel discussion not included in reference 29, and that these manuscripts well complement each other.

Comment on the relation with ref. 29:

Thank you for this detailed review on the relation between this manuscript and ref 29 (now ref 35), we fully agree with your comments and conclusions. We have added some additional words from our point of view here:

This manuscript puts the entire focus on the analysis of the measurements made by the several instruments on BepiColombo. Ref 35 (Aizawa, Persson et al., 2022) is focused on the global hybrid simulation, and, among other things, the purpose of that paper was to show how well the global hybrid model works, especially for this BepiColombo 2nd Venus flyby. This is a task that was very important to do, in particular for this manuscript, as we rely on the results of the simulation to help us confirm our interpretations about the measurements. Ref. 35 do not make any analysis of the measurements, which was done on purpose in order to leave that analysis for this manuscript alone. Therefore, ref 35 complements this manuscript very well, but it does not remove any novelty of this manuscript.

I do have several minor clarification questions that I believe should be addressed prior to any publication; these are outlined below.

General comment on Figure 3 and accompanying text (lines ~82-141): I found the descriptions and walk through of this figure, and the instrument observation strengths and limitations, to be very thorough and useful.

Thank you!

Line 55: is “peculiarities” the right word here? Can you mention what these are if so? Would “unique characteristics”, or something similar, be more appropriate?

Changed “peculiarities” to “unique characteristics”.

Line 67: While PVO ion measurements were limited in some properties, they still provided the fluid characteristics of the Venus environment. I suggest modifying this sentence to something like “did not have enough time resolution and energy range to provide comprehensive/complete particle characteristics”.

Good point! We agree that it is important to give credit to the great measurements and work made by the older missions and their teams. Added the word “comprehensive” to clarify this.

Line 86: Should the Mercury Magnetosphere Orbiter have the acronym “MMO”, not Mio?

Indeed, the acronym for this spacecraft would be MMO, but a tradition at JAXA is to change the name of their spacecraft after launch. MMO was thus re-named to Mio, which is why we used it in the manuscript. See ref. 16: Murakami et al., 2020. We added a note on this in the text to clarify.

Lines 144-159: discussion of the stagnation region as observed by the particle instruments: I found this section difficult to follow, and I suggest rewording it if possible. Can you clarify, is it that in Figure 5B, the ion distribution function (idf) is broad and so you are assuming that the ions have been heated and the flow stagnates? To my uninitiated eye, panels B and C seem quite similar – both have broad idfs. The main difference appears to be that in B, the counts do not overlap with the blue dot, while they do in C. Is this important? Perhaps you can show some sort of line plot with normalized integrated counts as a function of radius in FOV, to show that the idf in B is the broader than in C, if this is the case?

We have tried to reword the text and expand upon the descriptions a bit to hopefully bring some more clarity to what we want to say with this figure. We want to thank the reviewer for this comment as it made us realise an error in the label of the colorbar, which should be log normalised but was presented as linearly normalised. This means that the difference between the colours is larger than they first appeared. Another important note is that the sensitivity of the MIPA instrument is low, as it was designed for the higher fluxes present at Mercury and not the lower fluxes of the Venusian magnetosheath.

The main difference between the panels that we want to point out (which we tried to include better in the text now) is that the total counts is relatively smaller in B than C, and that it is relatively more focused in C than B. These both point to the difference in bulk speed and temperature in the two regions. With a high temperature and low bulk speed (5B) the phase space density “per bin” will be low, and so it will come close to the sensitivity limit of MIPA, thus having a low count in almost all pixels. With a higher bulk speed and lower temperature (5C) the phase space density “per bin” will be higher and thus the flow will be more focused and provide a higher count in the “interesting pixels” of MIPA, i.e. close to the expected bulk flow direction from the LatHyS simulation. We have included the bulk speed direction and the width of the proton distribution predicted for these regions by the LatHyS simulations (blue and grey dots respectively), which are well constrained by the upstream Solo solar wind measurements. The results from the LatHyS simulation also show this difference of a higher temperature for B than C, and a lower bulk speed in B compared to C.

Line 165: Given the field of view limitations discussed previously, what is the uncertainty here that the flow is ~aligned with the obstacle boundary? In Figure 5C, is there a symbol denoting the obstacle boundary? How is the obstacle boundary defined?

We have not shown the direction of the obstacle boundary in this figure as it could complicate things further. However, the measurements of MIPA show a strong peak within its FoV, and not at the edge, in a region where we do not expect to have several different distributions (see e.g. Spreiter and Stahara, ref 7-8). The peak is also located within 20-30° of the bulk flow direction from the LatHyS global hybrid simulations (blue dot), and the measured and simulation distributions are significantly overlapping. The bulk flow direction found by the simulation results (blue dot) are also shown in Fig 6A, where the flow is shown to be “along the boundary”, i.e. what the solar wind sees as the obstacle, which is the Venusian induced magnetosphere (added this to the main text). The location of the outer boundary of the induced magnetosphere, usually referred to as induced magnetosphere boundary or magnetic pileup boundary depending on the physics focused on, is well-established by several previous statistical studies (e.g. Martinecz et al., 2008; Zhang et al., 1991). Therefore, these results support that the flow is in the expected direction of the magnetosheath, along the obstacle boundary. It is important also to note here that Figure 5A-C are in the instrument frame and not fixed in Venus space, which can be seen by the

movement of the yellow dot (Venus' centre) in Fig 5A-C. Therefore, the directions of each pixel with respect to Venus will be different for each of the three panels. We hope that our edits of the manuscript related to Figure 5, in both the paragraph on the stagnation region and in the figure text, have helped clarify this.

Line 167: Text on lines 163-164 states that electron and ion energies decrease, but at line 167 the text states that the flow transitions from subsonic to supersonic. Are these consistent?

The subsonic vs supersonic regime is connected to the bulk flow speed vs the magnetosonic wave speed and is thus not only related to the change in average measured energy of the plasma. What we do see is a decrease in temperature, which we interpret as related to the change in regime where the flow (if we follow along the streamlines) is becoming less stagnant (i.e. decrease in temperature seen from both the FoV of MIPA and from the spread in energy) and starts to have a flow aligned with the obstacle boundary (observed from the FoV of MIPA as described in the comment above this). These are changes expected to happen as we go from a subsonic to supersonic regime, i.e. crossing the sonic line. We have changed the text to reflect this.

Line 168: “beyond which the plasma flow reverts to almost solar wind speeds further downstream” Is this the black line in Figure 3E? I suggest pointing that out if so. Can you comment on the accuracy of this interpretation, given the field of view issues discussed earlier?

Changed the text to “beyond which the plasma flow *is expected* to revert to almost solar wind speeds further downstream”, as we are referencing the results from the model by Spreiter and Stahara (ref 7-8). We are talking about the measured average ion energy and its comparison with the solar wind energy in a later paragraph, where we are referencing Figure 3E. However, note that MIA can measure the protons energy in the magnetosheath because the protons have a larger temperature there than in the solar wind. The proton energy can thus be confidently measured by MIA even with a limited FoV. Therefore, the comparison between MIA and PAS energy can tell us that the proton speed is reverted to almost solar wind energies this far from Venus in the magnetosheath.

Lines 175: At Mars, when spacecraft enter the MPB, there is sudden reduction in high energy (>~70 eV) electrons - that does not seem to be the case here (e.g. Vignes+ 2000). There is also no clear large scale rotation in magnetic field to a draped configuration. Do you still think that this is the MPB ?

Indeed, Vignes+2000 note a decrease in electron fluxes of energies >10 eV when crossing the MPB, together with a sharp increase in magnetic field magnitude. In agreement with them, we observe a decrease in the electron fluxes. However, Bertucci+03 noted that the MPB does not always show a sharp rise in the magnetic field at Venus, and that there is generally a decrease in the magnetic field fluctuations when entering the MPB. The observations by Bertucci+03 strengthens our interpretations that we have entered the MPB. However, similarly to the IMB discussion a few lines down, we do not think BepiColombo crossed the MPB completely, and most likely only skimmed this region, which can be several 100 km thick (ref 25-27). We have updated the text to be clearer, and included the Vignes and Bertucci references.

Line 202: Is the word “instantaneous” appropriate here? What is the sampling instantaneous with?

This wording is indeed a bit ambiguous. The word was chosen to indicate that the full magnetosheath tour of BepiColombo happens in a very short time and that the solar wind was remarkably stable during this time, thus indicating that we get a chance of seeing the spatial structures without the interference of large temporal structures. However, we acknowledge that

this might not be understood from the current wording, so we removed “almost instantaneous”, and instead added a sentence after that specifically points out this important information.

Line 212: Would the word “interpretations” be better than “assumptions”?

Agreed! Changed.

Figure 2 caption: Are all of the acronyms defined in the text?

Yes, we checked that all acronyms are defined in the manuscript text and added a note on this in the caption.

Figure 3: does panel B show data for all ion species, or just a particular species? Please clarify this in the manuscript if it’s not done so already.

Added a note in Figure 3 caption that “All ion measurements are shown integrated over mass.”

Figure 5: this ties in with my comments about lines 144-159 above. For panel B, is the blue dot where the stagnation region is expected to be (the blue dots are labeled as the expected average bulk flow velocity in the caption)? If so, why are there no counts in the sector overlapping the blue dot, if this is the stagnation region?

We have edited both the main text and the figure text of Figure 5 to try to make it clearer exactly what we want to share with this figure. The blue dot is from the LatHyS hybrid model and shows the expected flow direction from the simulations. The large temperature of the simulation results in panel B (grey dots) show that the bulk flow direction is not as important as the spread in itself. In addition, as we mentioned above, the sensitivity of the MIPA instrument together with the lower phase space density for a low bulk speed + high temperature flow is the main information we want to take out from this figure. Text is edited to be clearer about this point.

References:

EG Vignes, D., Mazelle, C., Reme, H., Acuña, M. H., Connerney, J. E. P., Lin, R. P., ... & Ness, N. F. (2000). The solar wind interaction with Mars: Locations and shapes of the bow shock and the magnetic pile-up boundary from the observations of the MAG/ER Experiment onboard Mars Global Surveyor. *Geophysical Research Letters*, 27(1), 49-52.

REVIEWER #3 (REMARKS TO THE AUTHOR):

Review by T. E. Cravens

This paper presents a wealth of data from BepiColombo’s second encounter with Venus and from the Solar Orbiter upstream of Venus in the solar wind. The paper focuses on the plasma and field properties in the region downstream of the bow shock including the magnetosheath and stagnation regions. Although these regions were studied earlier by the Pioneer Venus and Venus Express missions, the current paper provides a different and more coherent view of these regions. Results from global hybrid simulations of the solar wind interaction with Venus (also shown in Aizawa et al., 2022)

are very helpful in putting the BepiColombo data into context. The paper will make an important contribution to our understanding of how the solar wind interacts with non-magnetic planets. I recommend publication with a few rather minor suggestions for improvement as listed below.

Dear Prof. Cravens, thank you for reviewing our manuscript and providing positive and constructive comments. We have addressed your three comments point-by-point below.

- Three regions need to be more carefully defined and distinguished in the paper: magnetosheath, stagnation region, and magnetic pile-up region. Looking at Figures 1 and 3, the brown region labeled stagnation region appears to be subsolar magnetosheath plasma downstream of the shock. And the flow continues to be slow into the purple region where the magnetic field is increasing (magnetic pile-up region).

We have edited the text to be clearer that the stagnation region and the sonic line are both part of the subsolar magnetosheath. We have also adjusted the explanation of the magnetic pileup region and how it is related to these observations.

- The data is clearly presented in the paper, despite its complexity. One possible exception is Figure 5 which shows MIP and MIPA proton distributions. It is difficult to explain such complex data from such instruments in a rather short paper. A few more sentences might help.

Thank you! Indeed Figure 5 was noted to need better a description from all reviewers. We have updated the text and included some additional sentences to hopefully make it easier to understand.

- Lines 234 to 255 in the Discussion section puts the data into a broader dynamical context. The statement on magnetic pressure in lines 240-244 can be made even more explicit by stating the magnetic pressure (i.e., 1.2 nPa for 55 nT) and upstream solar wind dynamic pressure $\approx ?$ from Solar Orbiter. A general statement might also help the reader: solar wind dynamic pressure tends to be converted into thermal pressure (shock and stagnation) and then into magnetic pressure (barrier/pile-up region).

Yes, we agree. We have included the calculations made to find the presented numbers in the manuscript and included a general statement of this pressure balance as suggested.

REVIEWERS' COMMENTS

Reviewer #1 (Remarks to the Author):

The authors have responded well to the comments and the paper is now suitable for publication

Reviewer #2 (Remarks to the Author):

Thank you to the authors for providing detailed responses to my questions. These have been addressed, and I believe the manuscript is suitable for publication.

Reviewer #3 (Remarks to the Author):

The authors have done a thorough job of responding to the reviewer comments. I think that the revised paper is now acceptable.